# Evolving life-history traits promote biodiversity via eco-evolutionary feedback mechanisms

P. Catalina Chaparro-Pedraza[1,2,3]*, Claudia Bank[2,3]

**1** Department Systems Analysis, Integrated Assessment and Modelling, Swiss Federal Institute of Aquatic Science and Technology (EAWAG), Dübendorf, Switzerland, **2** Institute of Ecology and Evolution, University of Bern, Bern, Switzerland, **3** Swiss Institute of Bioinformatics, Lausanne, Switzerland

* Catalina.Chaparro@eawag.ch

## Abstract

To what extent is biodiversity shaped by environmental conditions, and to what extent is it the result of self-organization? Both natural processes and organismal properties may contribute to promoting diversity. Here, we show that one such process, namely natural selection, and an organismal property, namely life history, interact in a feedback mechanism that promotes the emergence of diversity. We illustrate how this mechanism operates using various models of ecological diversification driven by intraspecific resource competition, in which both a niche trait that determines resource use and a life history trait can evolve. We find that natural selection acting on life history traits leads to increased competition, which, in the presence of ecological opportunity, facilitates niche diversification. As a consequence, the environmental conditions for diversification are more restrictive in the absence of life history evolution than in its presence. Our findings indicate a strong influence of life history evolution on ecological processes that in turn shape the origin of biodiversity. Our results call for a better integration of life history evolution and niche diversification in both theoretical and empirical realms.

## Introduction

Ecosystems are complex adaptive systems in which system properties such as species diversity and community structure emerge from interactions among components. Such systems exhibit a large extent of self-organization based on a hierarchy of feedback mechanisms in which the outcome of interactions promotes further interactions [1–4]. Although there is growing awareness that feedback mechanisms are key to the maintenance of biodiversity [5,6], their role in the emergence of biodiversity is unknown.

The extraordinary diversity of life on Earth has emerged through a variety of biological processes. One of these is ecological diversification, during which a single ancestral population diversifies into ecologically different species that exploit a variety

provided the original author and source are credited.

**Data availability statement:** The code implemented for this study is available in the zenodo repository with DOI:https://doi.org/10.5281/zenodo.17049771.

**Funding:** This work was supported by the H2020 European Research Council (Grant No. 804569 to CB https://cordis.europa.eu/project/id/804569). The funder had no role in study design, data collection and analysis, decision to publish, or preparation of the manuscript.

**Competing interests:** The authors have declared that no competing interests exist.

of niches [7,8]. This process has produced diverse clades through rapid diversification, including Darwin's finches [9], Caribbean anole lizards [10], and cichlid fishes [11]. In these and other examples of ecological diversification, the availability of relatively unexploited ecological niches, known as ecological opportunity, results in a regime of frequency-dependent selection that emerges from competition. In such a regime, intermediate phenotypes have a fitness disadvantage compared with more extreme phenotypes, causing phenotypic diversification, which may ultimately result in speciation in case that reproductive isolation evolves between populations with divergent phenotypes [12,13].

Competition among individuals is one of the main processes driving ecological diversification [14–16]; therefore, any factor modulating the strength of intraspecific competition influences the occurrence of ecological diversification. Much research has investigated how favorable environmental conditions, such as those causing high productivity or low mortality, facilitate ecological diversification by strengthening intraspecific competition [17–20]. However, the possibility that evolutionary processes, such as natural selection, alter intraspecific competition and thereby influence diversification has not been explored. This is because existing diversification models and empirical research have mostly focused on the evolution of traits controlling resource acquisition, whereas other traits received little attention. If natural selection, by driving changes in such traits, strengthens intraspecific competition, it could facilitate diversification. However, it is unknown whether natural selection tends to favor conditions that promote diversification and, thus, high diversity [8].

A key insight into this problem comes from previous theoretical work on life history evolution showing that the evolution of life history traits can alter intraspecific competition [21]. Life history traits such as the time organisms take to grow, when they become mature, and how many offspring of a particular size they produce, influence their reproductive success. Natural selection can drive changes in these life history traits, shaping the life cycle of organisms to increase their fitness in the face of ecological challenges posed by the environment [22–24]. After colonizing a novel habitat, natural selection may drive life history adaptation in a population by replacing life history phenotypes of low reproductive success with phenotypes of higher success. This gradual increase in reproductive success results in stronger intraspecific competition [21], potentially facilitating diversification. To date, we do not know how life history evolution, by affecting intraspecific competition, may alter ecological diversification driven by niche specialization.

Here, we identify life history evolution as a general facilitator of ecological diversification by analyzing three fundamentally different theoretical models. Each model considers ecological niches representing ecological opportunity and a niche trait that determines resource use. (Fig 1A). The models differ in the description of the population dynamics, the evolving life history trait (Fig 1B), and other assumptions underlying organismal processes (e.g., a linear functional response describes individual feeding rate in Model 1 and Model 3 versus a saturating functional response in Model 2). We thus demonstrate that the facilitating effect of life history evolution on diversification holds beyond specific model assumptions and life history traits. Our results thus provide a robust quantitative link between life history evolution and trophic diversification.

## A) Ecological opportunity

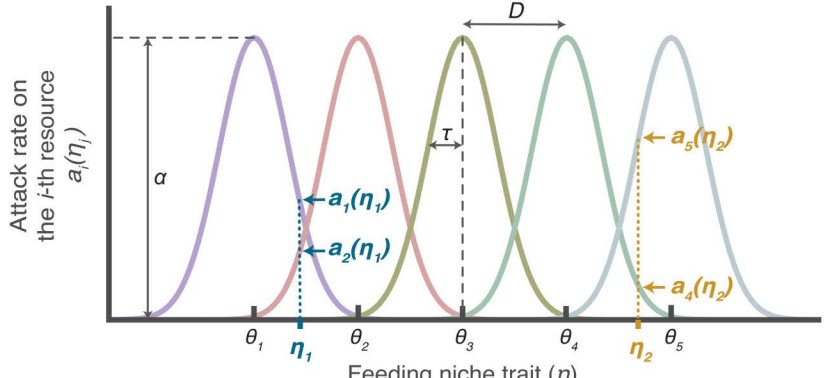
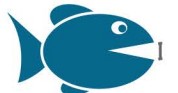
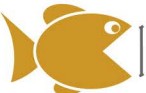

Feeding niche trait $\eta_1$
(e.g. maximum gape)

Feeding niche trait $\eta_2$
(e.g. maximum gape)

Blue: An organism with feeding niche trait $\eta_1$ has an attack rate on resource 1 and resource 2, and its attack rate on other resources is nearly zero. Yellow: An organism with niche trait $\eta_2$ feeds mostly on resource 4 and resource 5, and its attack rate on other resources is nearly zero.

## B) Population dynamics

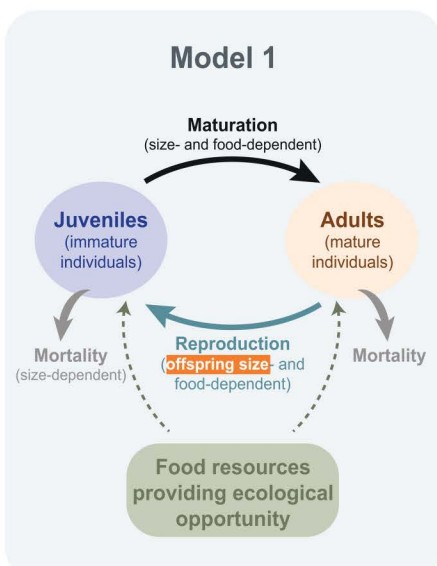
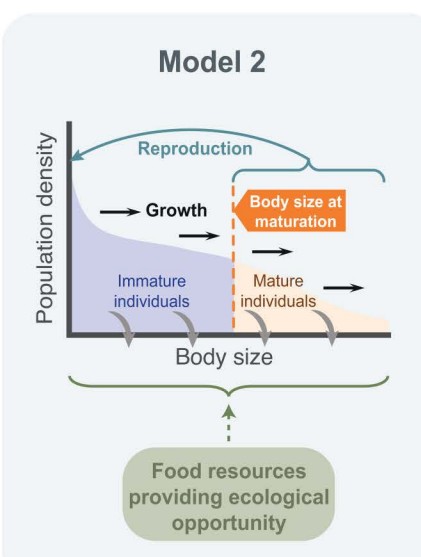
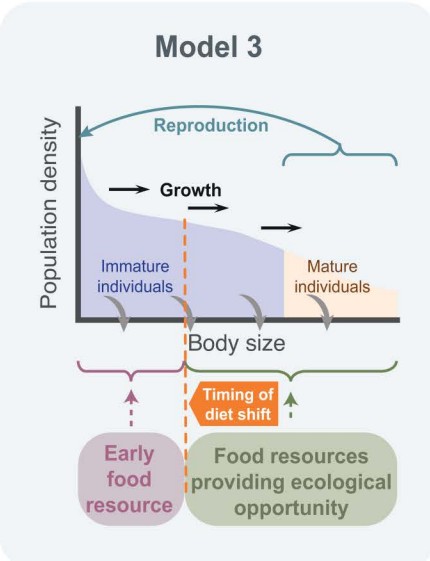

**Fig 1. Structure of the models. A)** Our models consider the existence of multiple ecological niches, or rather the resources that form the niches, prior to the arrival of an ancestral population. To consume each resource, there exists an optimal feeding niche trait $\theta_i$. Organisms differ in the feeding niche trait (e.g., the maximum gape in fish or reptiles, or the bill size in birds, which determines the size of the food particles that they can ingest). **B)** We study three fundamentally different models. In Model 1, the population is structured by two life stages: juveniles and adults; whereas in Models 2 and 3, the population is structured by body size, which changes over an individual's lifetime. The models also differ in the evolving life history trait, indicated in the orange boxes. Organismal processes such as reproduction, mortality, maturation (in Model 1), and individual growth (in Models 2 and 3) are represented by solid arrows. Trophic interactions between resources and consumers are represented by dashed arrows. In Model 2 and Model 3, the gray curved arrows correspond to the population losses caused by mortality, and the black arrows to individual growth. In Model 3, competition for food and access to ecological opportunity occurs only after the timing of the diet shift (organisms do not compete for the early food resource).7.

## Models and results

We formulate and analyze three alternative models to examine how natural selection on life history traits influences ecological diversification. The core of each model is the description of various organismal processes, that is, feeding, growth,

reproduction, and mortality, as a function of the environment (food availability) and the organism itself. The ecological dynamics emerging from this description determine the local fitness landscape and, thus, the evolution of life history and feeding niche. Applying adaptive dynamics techniques [25,26], we identify the conditions that enable diversification (analytically derived for Model 1) and determine the evolutionary trajectories of diversifying lineages.

## Ecological opportunity and food resource use in the three models

Building on existing theory on ecological diversification [17,27–29], our models consider $n$ food resources with density $F_i$ ($i = 1, \ldots, n$), and $m$ emerging consumer ecomorphs ($j = 1, \ldots, m$), each with life history trait $\ell_j$ and feeding niche trait $\eta_j$ determining their resource use. In the absence of consumers, resource density dynamics follow $\frac{dF_i}{dt} = \rho (F_{i\,max} - F_i)$, where $\rho$ and $F_{i\,max}$ are the renewal rate and the carrying capacity of the $i$th resource, respectively. Because the product of the carrying capacity and the supply rate corresponds to the maximum possible production rate of each resource, we measure the total productivity of food resources as $P = \rho \sum_i F_{i\,max}$.

Assuming that there exists an optimal trait value $\theta_i$ to consume each resource, these optimal traits are ordered along a one-dimensional niche trait space (i.e., $\theta_1 < \theta_2 < \ldots < \theta_n$), separated by a distance $D > 0$. The attack rate, $a_i(\eta_j)$, of an ecomorph with feeding niche trait $\eta_j$ on the $i$th resource equals the maximum attack rate $\alpha$ when its feeding niche trait $\eta_j$ equals $\theta_i$, and decreases in a Gaussian manner as $\eta_j$ moves away from $\theta_i$, that is $a_i(\eta_j) = \alpha \exp\left[-(\eta_j - \theta_i)^2 / (2\tau^2)\right]$ (Fig 1A). In this expression, $\tau$ determines the width of the Gaussian function and thus the breadth of the feeding niche. Hence, when $\tau$ is small, an ecomorph must be highly specialized to successfully attack resource $i$. Consequently, a tradeoff between feeding on alternative food resources emerges, in which the specialization on one food resource is at the expense of specialization on the others [30]. Such tradeoffs have been generally observed in heterotrophic organisms, including bacteria [31], insects [32], and vertebrates [33].

## Model 1: Diversification and evolution of offspring size

Model 1 considers an ecomorph population $j$ that is composed of juvenile and adult individuals. Individuals are born at size $\ell_j$ and mature (i.e., enter the adult stage) when they reach size $w$. The feeding rate of juveniles of the $j$th ecomorph is $c_{J,j}(\eta_j, F) = \gamma \sum_{i=1}^{n} a_i(\eta_j) F_i$ and of adults is $c_{A,j}(\eta_j, F) = \sum_{i=1}^{n} a_i(\eta_j) F_i$, with $F = (F_1 \ldots F_n)$. The general rule across animal taxa is that larger organisms have higher foraging capacity than smaller conspecifics [34–36]. To meet this rule in the model, the factor $\gamma < 1$ scales the juvenile feeding rate, such that it is lower than that of the adults. The flux of energy that juveniles allocate to somatic growth is $\phi_j(\eta_j, F) = \max(\varepsilon c_{J,j}(\eta_j, F) - \nu, 0)$, where $\varepsilon$ is the assimilation efficiency and $\nu$ is the metabolic cost. Hence, juveniles grow when the assimilated food exceeds the metabolic cost $\nu$ and do not shrink in size. De Roos and Persson showed that the maturation rate depends on the somatic growth, the mortality rate, and the size range over which an individual grows in the juvenile stage (see Box 3.1 in ref [37]), as given by:

$$\Phi_j(\eta_j, \ell_j, F) = \begin{cases} \dfrac{\phi_j(\eta_j, F) - \delta_J}{1 - \left(\frac{\ell_j}{w}\right)^{1 - \frac{\delta_J}{\phi_j(\eta_j, F)}}} & \text{if } \phi_j(\eta_j, F) > \delta_J \\ 0 & \text{otherwise,} \end{cases}$$

where $\delta_j$ is the juvenile mortality rate. The maturation rate thus decreases as the range over which an individual grows in the juvenile stage increases; that is, as the ratio $\frac{\ell_j}{w}$ decreases. Conversely, the maturation rate increases as the rate at which individuals grow increases. Adults only reproduce when assimilated food exceeds the metabolic cost incurred by adults, $\nu$, such that their birth rate is $\beta_j(\eta_j, \ell_j, F) = \max(\varepsilon c_{A,j}(\eta_j, F) - \nu, 0)/\ell_j$. Birth rate is divided by the offspring size, reflecting the well-known tradeoff between offspring number and size that has been ubiquitously observed across diverse plant and animal taxa [22,38–40]. Additionally, juvenile survival was documented to increase with offspring size in wild populations of diverse animal species [41]. Because larger organisms often have greater survival than smaller

conspecifics [42–53] and this relationship scales exponentially with respect to body size [54], we assume that juvenile mortality decreases with offspring size, following $\delta_J(\ell_j) = \delta_{max}\exp(-\ell_j)$, where $\delta_{max}$ is a boundary of maximum possible mortality. Adults die at a rate $\delta_A$. Starvation mortality is not considered because, at the ecological equilibrium, a population is viable (i.e., its density is positive) only if starvation conditions do not occur, i.e., if $\varepsilon c_{J,j} > \nu$ and $\varepsilon c_{A,j} > \nu$.

Based on the assumptions described above, the dynamics of juveniles $J_j$ and adults $A_j$ of the $j$th ecomorph, as well as of food resources, follow

$$\frac{dJ_j}{dt} = \beta_j(\eta_j, \ell_j, F) A_j - \Phi_j(\eta_j, \ell_j, F) J_j - \delta_J(\ell_j) J_j$$

$$\frac{dA_j}{dt} = \Phi_j(\eta_j, \ell_j, F) J_j - \delta_A A_j$$

$$\frac{dF_i}{dt} = \rho(F_{i\ max} - F_i) - \gamma \sum_{j=1}^{m} a_i(\eta_j) F_i J_j - \sum_{j=1}^{m} a_i(\eta_j) F_i A_j \tag{1}$$

The total population density, $N_j = A_j + J_j$, varies following

$$\frac{dN_j}{dt} = [\beta_j(\eta_j, \ell_j, F) C_j - (\delta_J(\ell_j)(1 - C_j) + \delta_A C_j)] N_j, \tag{2}$$

where $C_j$ is the fraction of adults (for the ecological dynamics rewritten in terms of $N_j$ and $C_j$, see eq. S1.1 in S1 Text). The per capita growth rate, and thus the fitness of the $j$th ecomorph, is $W_j = \frac{1}{N_j}\frac{dN_j}{dt}$. Using this fitness expression and following adaptive dynamics techniques [25,26], we obtained analytical expressions for the selection gradient and the curvature of the fitness landscape when both the feeding niche trait and the offspring size evolve (see S1 Text: eq. S1.4–S1.6). For a population encountering an environment with two different food resources with carrying capacities $F_{1\ max} = F_{2\ max}$, we obtained analytical expressions for the food and population densities as well as the fraction of adults at the ecological equilibrium (see S2 Text). Using these quantities and the expressions for the selection gradient and the curvature of the fitness landscape, we analytically derived the conditions that enable diversification (see S3 Text) and determined their relationship with the life history trait (see S4 Text).

Additionally, we numerically computed the evolutionary trajectories of a diversifying lineage that encounters an environment with ecological opportunity represented by two or more different food resources. To do so, we used eq. S1.1–S1.3 (S1 Text) and parameter values in Table A in S5 Text. Moreover, we evaluated whether a diversification event occurs by calculating the curvature of the fitness landscape (using eq. S1.6 in S1 Text) when directional selection in the feeding niche trait ceases (i.e., when the change in this trait was smaller than 1E−8 per evolutionary time step). If the curvature of the fitness landscape indicated that the population experiences disruptive selection in the feeding niche trait (selection in the life history trait is always directional or stabilizing), population $j$ was split into two distinct ecomorph populations with trait values 0.1% larger and smaller than $\eta_j$, and abundance equal to $N_j/2$. Subsequently, the evolutionary trajectories were computed simultaneously for the two ecomorph populations until directional selection in the feeding niche trait ceased, and the curvature of the local fitness landscape was evaluated for each ecomorph. Each computation finished when all ecomorph populations experienced stabilizing selection. Throughout each computation, we followed the number of ecomorphs, their traits, and their densities, as well as the densities of resources, which we used to calculate the fitness gradient and the curvature of the fitness landscape (using eq. S1.4–S1.6 in S1 Text).

## By altering the stability of evolutionary equilibria, life history influences the conditions for diversification

For diversification to occur, theory proposes that a population must experience disruptive selection for a significant amount of time [12]. This is possible when natural selection drives a population towards a local minimum of the fitness landscape, where intermediate phenotypes have a fitness disadvantage relative to more extreme phenotypes [13,55]. Although our model describes the evolution of two phenotypic traits, ecological opportunity is provided only along the axis of the feeding niche trait. As a result, the possibility of a local fitness minimum exists only along this trait axis. Consequently, our model behaves similarly to previous diversification models with a single evolving trait (i.e., one-dimensional trait space diversification models, e.g., ref [55,56]). In such models, two conditions need to be satisfied for diversification to occur in a population colonizing an environment with two different food resources:

Condition 1 (condition for mutual invasibility according to Geritz and colleagues [55]): The mean phenotype of the population at an evolutionary equilibrium must be a local fitness minimum. A population satisfies this condition when:

$$\frac{D}{\tau} > 2. \tag{3}$$

(see analytical derivation in S3 Text: eq. S3.19). Because the ratio $D/\tau$ determines the strength of the tradeoff between feeding on the alternative food resources, eq. 3 implies that this tradeoff needs to be sufficiently strong to induce disruptive selection [27,57]. In other words, the broader the niche of a population, i.e., the larger $\tau$, the greater must be the distance between the optimal trait values for feeding on alternative food resources for disruptive selection to occur. Otherwise, the generalist strategy, corresponding to the evolutionary equilibrium in between the optima to feed on the resources, is a fitness maximum and thus selection is stabilizing.

Condition 2 (condition for convergence stability according to Geritz and colleagues [55]): The evolutionary equilibrium at which the population experiences disruptive selection must be an attractor of the evolutionary dynamics, or in other words, natural selection must be able to drive a population toward this evolutionary equilibrium. This condition is satisfied when productivity exceeds the threshold

$$P > \frac{\rho G D^2}{4 E \tau^2 \alpha} e^{\frac{D^2}{8\tau^2}}, \tag{4}$$

where $E = \varepsilon C/\ell$ is a scaling factor of the population birth rate and $G = vC/\ell + \delta_A C + \delta_J(\ell)(1 - C)$ is the rate of biomass loss through various processes (e.g., metabolic maintenance, mortality). These two compound variables regulate population growth, as shown by rewriting eq. 2 as $\frac{dN_j}{dt} = \left[ E \left( \sum_{i=1}^{n} a_i (\eta_j) F_i \right) - G \right] N_j$. Population growth increases as $E$ increases due to greater efficiency to assimilate food (high $\varepsilon$) or a higher fraction of adults in the population (high $C$), or due to smaller offspring size (small $\ell$) that increases the adult birth rate. Conversely, population growth decreases as $G$ increases due to increased metabolic cost (high $v$), higher adult mortality (high $\delta_A$), or higher juvenile mortality (high $\delta_J$).

Eq. 4 implies that there exists a minimum productivity of the environment that enables the evolutionary equilibrium at which the population experiences disruptive selection to be an attractor of the evolutionary dynamics (see analytical derivation in S3 Text: eq. S3.21). Below this threshold of minimum productivity, the population density is low and intraspecific competition is weak, hindering diversification. This threshold depends on parameters underlying the provision of ecological opportunity, such as the distance between the optimal values to feed on alternative resources $D$ and the resource supply rate $\rho$, as well as organismal properties, such as the maximum attack rate $\alpha$, and the breadth of the feeding niche $\tau$. Additionally, this threshold also depends on the compound variables, $E$ and $G$. Large values of $E$ or small values of $G$ reduce this threshold, enabling diversification over a larger range of productivities. This is because by increasing $E$, the population birth rate increases. Analogously, by reducing $G$, the rate of biomass loss decreases. Consequently, in both cases, the population density increases, leading to stronger intraspecific competition that, in turn, facilitates diversification.

Life-history evolution through changes in offspring size in Model 1 does not affect Condition 1, but it affects Condition 2. Indeed, eq. 3 shows that the breadth of the feeding niche, $\tau$, is the only organismal property influencing Condition 1. In contrast, multiple organismal properties, including offspring size, influence Condition 2. As a result, changes in this life history trait driven by natural selection alter the stability condition that determines when selection drives the feeding niche trait to the value where it becomes disruptive. This is due to the effect of offspring size evolution on the population birth rate through $E$ and the biomass losses through $G$. These effects are direct; for instance, the offspring size negatively affects the birth rate, i.e., $E$ decreases as $\ell$ increases. Similarly, because $\delta_J(\ell)$ decreases with increasing $\ell$, the offspring size negatively affects the mortality experienced by the offspring, and thus the biomass losses $G$. In addition, these effects are indirect, mediated by changes in population composition, $C$. Hence, by influencing the population composition, its birth rate, and its losses, the offspring size determines whether the feeding niche trait value at which selection is disruptive is an attractor or repeller of the evolutionary dynamics and thus dictates whether diversification occurs or not.

## The evolution of life history traits enables diversification

To understand how life history evolution influences ecological diversification, we numerically computed evolutionary trajectories in two scenarios: (1) a null model in which only the niche trait can evolve (i.e., the mutation rate for offspring size is set to zero), and (2) an alternative model in which both traits, niche and offspring size, can evolve. In the first scenario, in which offspring size remains constant (Fig 2A), the niche trait of the ancestral population evolves towards one of the resource feeding optima (Fig 2B). At this point, diversification cannot occur because selection is stabilizing. In the second scenario, the niche trait is initially driven toward one of the resource feeding optima (between time 0 and 5,000 in the examples given here); however, later, when the offspring size approaches an intermediate trait value (offspring size of 1.5 in Fig 2C), the direction of selection on the niche trait changes. As a result, the niche trait evolves to the value between the two optima. At this point, selection becomes disruptive; therefore, the population undergoes a diversification event in the niche trait. After diversification, the niche traits of the two resulting ecomorph populations diverge. Each population then evolves towards its nearest optimal feeding trait value (Fig 2D).

To better understand the mechanism by which life history evolution enables diversification, we analyzed the fitness landscape of the niche trait experienced by the colonizing population. Our analysis revealed that ecological diversification can only occur when the offspring size is near its optimal value (purple vertical line in Fig 2E). In the neighborhood of the optimal offspring size, food resources are strongly depleted, causing strong intraspecific competition (Fig 2F). Under this regime of strong competition, the gain from becoming more specialized on the underused resource outweighs the loss from a decreased intake of the most used resource. As a consequence, the niche trait evolves in the direction that enables increasing use of the underused resource. This process ceases when both resources are equally exploited, that is when the niche trait value is equal to the intermediate value between the two optima to feed on the resources (niche trait 0.5 in Fig 2). Therefore, this niche trait value is an attractor of the evolutionary dynamics that may be either (1) a global attractor, and thus, evolution drives the niche trait towards it regardless of the initial trait value of the colonizing population (e.g., when offspring size is larger than 2.2. and smaller than 4.2 in Fig 2G), or (2) a local attractor, and thus, evolution can only drive the niche trait towards it if the trait value of the colonizing population is in its neighborhood (e.g., when offspring size is between 1.2 and 2.2 or between 4.2 and 5.9 in Fig 2G). In the first case, diversification always occurs, whereas in the second case, it is contingent on the initial niche trait of the colonizing population. Conversely, when the offspring size is far from its optimal value, the niche trait value between the two optima to feed on the resources is a repeller of the evolutionary dynamics, and consequently, evolution drives the niche trait away from this value (offspring size elow 1.2 and above 5.9 in Fig 2G).

## The evolutionary rate of life history traits influences diversification and the ensuing diversity

The effect of life history traits on the fitness landscape of the feeding niche trait suggests that the speed at which life history traits evolve affects diversification. To test this, we computed the evolutionary trajectories of two lineages colonizing

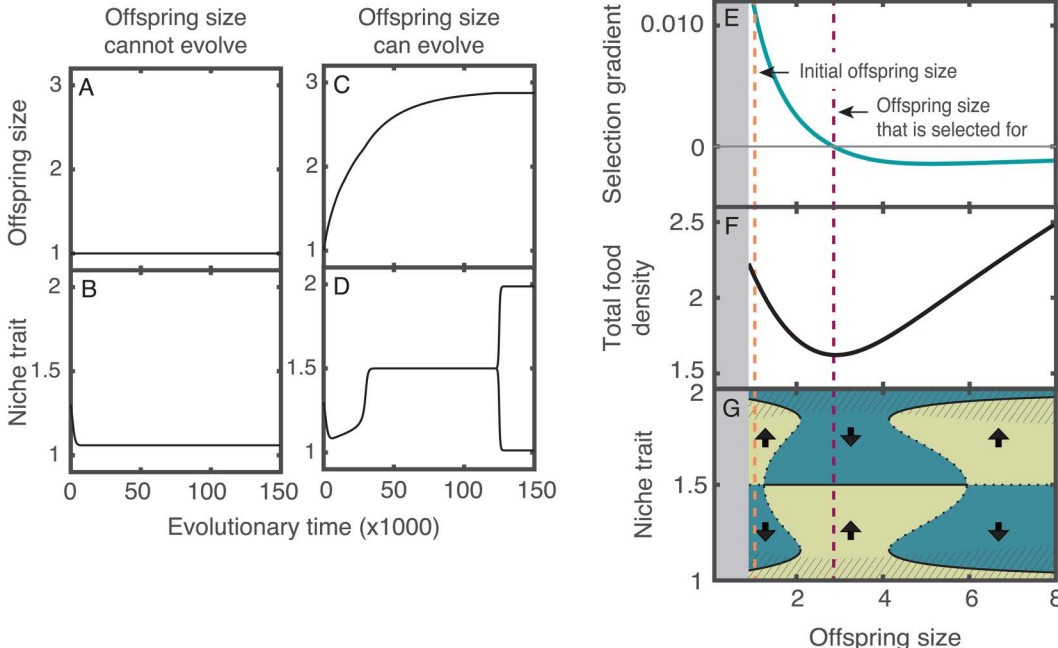

**Fig 2. The evolution of offspring size enables diversification.** Evolutionary dynamics of a population that colonizes an environment with two food resources (optimal niche traits to feed on resources are $\theta_1 = 1$, $\theta_2 = 2$) **A, B)** when the offspring size cannot evolve, and **C, D)** when it can evolve. **E)** Selection gradient with respect to the offspring size of the colonizing population. **F)** Sum of the food densities at the ecological equilibrium as a function of the offspring size. **G)** Fitness landscape of the niche trait as a function of the offspring size. Diversification occurs only when the offspring size of the population is near the optimal trait (indicated with purple vertical dashed line) because the niche trait evolves towards a fitness minimum, where the trait equals 1.5 (arrows indicate the direction of selection). Evolutionary equilibria (black lines; solid when stable and dotted when unstable) in the striped regions are fitness maxima, whereas outside these regions they are fitness minima. In the gray region in the left, the population is extinct. In **A–D**, the initial offspring size and niche trait are 1 and 1.3, respectively. In F, the niche trait value is fixed and equal to 1.4. However, regardless of the niche trait value, the food density is always depleted to the lowest level at the value of the offspring size that is selected for (analytical proof in S4 Text). Parameter values as in table A in S5 Text. The code needed to generate this Figure can be found in https://zenodo.org/records/17049771.

an environment with 10 different unexploited niches. These lineages differ in the mutation rate of the life history trait (i.e., the offspring size), thereby representing different levels of evolvability for this trait, but share the same mutation rate for the feeding niche trait. At time 0, both lineages are seeded at opposite ends of the trait space, ensuring equal access to the available niches. The computation shows that the diversification process initiates earlier in the lineage with the highest mutation rate (Fig 3). This is because, in this lineage, the rate of change of the offspring size is higher, enabling natural selection to drive this trait towards the optimal value faster. Then, in the neighborhood of the optimal offspring size, natural selection drives the feeding niche trait towards the value in between two optima (e.g., between $\theta_1$ and $\theta_2$ at time 2,500), where selection becomes disruptive, enabling diversification. Conversely, in the lineage with the lower mutation rate, the evolutionary process driving changes in the offspring size is slower. Consequently, it takes roughly twice as long for the feeding niche trait to reach the value in between two optima (e.g., between $\theta_9$ and $\theta_{10}$ at time 5,000). As a result of the earlier diversification process, the lineage with the higher evolutionary rate in the life history trait occupies more niches, resulting in higher diversity.

## Local adaptation facilitates diversification

We next explored how variation in environmental conditions across habitats affects life history evolution and diversification. To this end, we computed evolutionary trajectories for environments that differed in the mortality risk experienced by

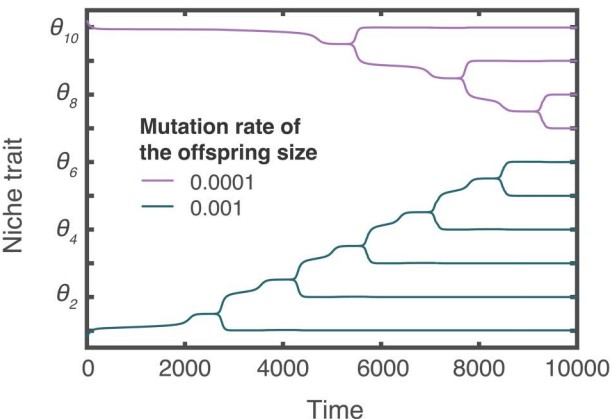

**Fig 3. The speed of offspring size evolution influences lineage diversity.** Evolutionary trajectories of two lineages colonizing an environment with 10 different unexploited food resources. The lineage with a higher mutation rate in offspring size results in 1.5-fold higher diversity than the lineage with lower mutation rate (the former diversifies into six different ecomorph populations, whereas the latter diversifies into four). The total productivity is 2 gL$^{-1}$ unit time$^{-1}$. The lineages are seeded at the beginning of the simulation in the extremes of the niche trait space (trait values of the ancestral populations are 0.8 and 10.2). The offspring size of both ancestral populations is 1. Parameter values as in table A in S5 Text. The code needed to generate this Figure can be found in https://zenodo.org/records/17049771.

juveniles. In more risky environments, natural selection favors a larger offspring size because juvenile mortality decreases as offspring size increases. The optimal offspring size is therefore shifted towards a larger trait value when the mortality risk of juveniles is higher (top row in Fig 4A). Interestingly, we observed that the range of productivity levels over which diversification can occur is larger when the offspring size is near its optimum than when it is far from the optimum (bottom row Fig 4A). Hence, by driving offspring size towards the optimal trait value, natural selection reduces the threshold of minimum productivity required for diversification to occur, facilitating diversification. We proved this effect analytically in S4 Text by showing that eq. 4 always assumes its minimum value when the offspring size is at its optimum.

## Model 2 and 3: Diversification and evolution of maturation size and timing of diet shift

To test the generality of our results, we examined how natural selection influences ecological diversification by driving changes in other life history traits (Fig 1B). In model 2, we studied the effect of the evolution of body size at maturation, which determines the transition of energy allocation from growth to reproduction (Fig A in S6 Text). Here, the optimal energy allocation strategy, and thus the optimal size at maturation, depends on the tradeoff between early reproductive investment and individual growth that increases size-dependent survival [24]. In this model, we studied evolutionary trajectories in alternative environments that differ in the magnitude of size-dependent survival to examine how environmental variation in survival affects the evolution of body size at maturation and diversification. In Model 3, we studied the consequences of the evolution of the timing of a diet shift, a key life history trait that determines when organisms with ontogenetic diet shifts begin exploiting a secondary food source. Such shifts are widespread across the animal kingdom, occurring in taxa ranging from amphibians and fishes to many invertebrates [58,59]. The timing of a diet shift involves a tradeoff between individual growth potential and survival [60,61], and is thus fundamental in determining individual fitness [ 61–64]. In this model, we studied evolutionary trajectories in alternative environments that differ in the mortality experienced by small individuals (i.e., individuals feeding on the early food resource) to examine how environmental variation in mortality in early life influences the evolution of the timing of a diet shift and diversification. For both models, we leveraged existing ecological models [65–67] that we extended to include evolution of the life history and feeding niche trait. See S6 Text for a detailed description of the two models. All analyses were performed using the PSPManalysis software package [68], which allows for the evolutionary analysis of population models structured by body size [69,70].

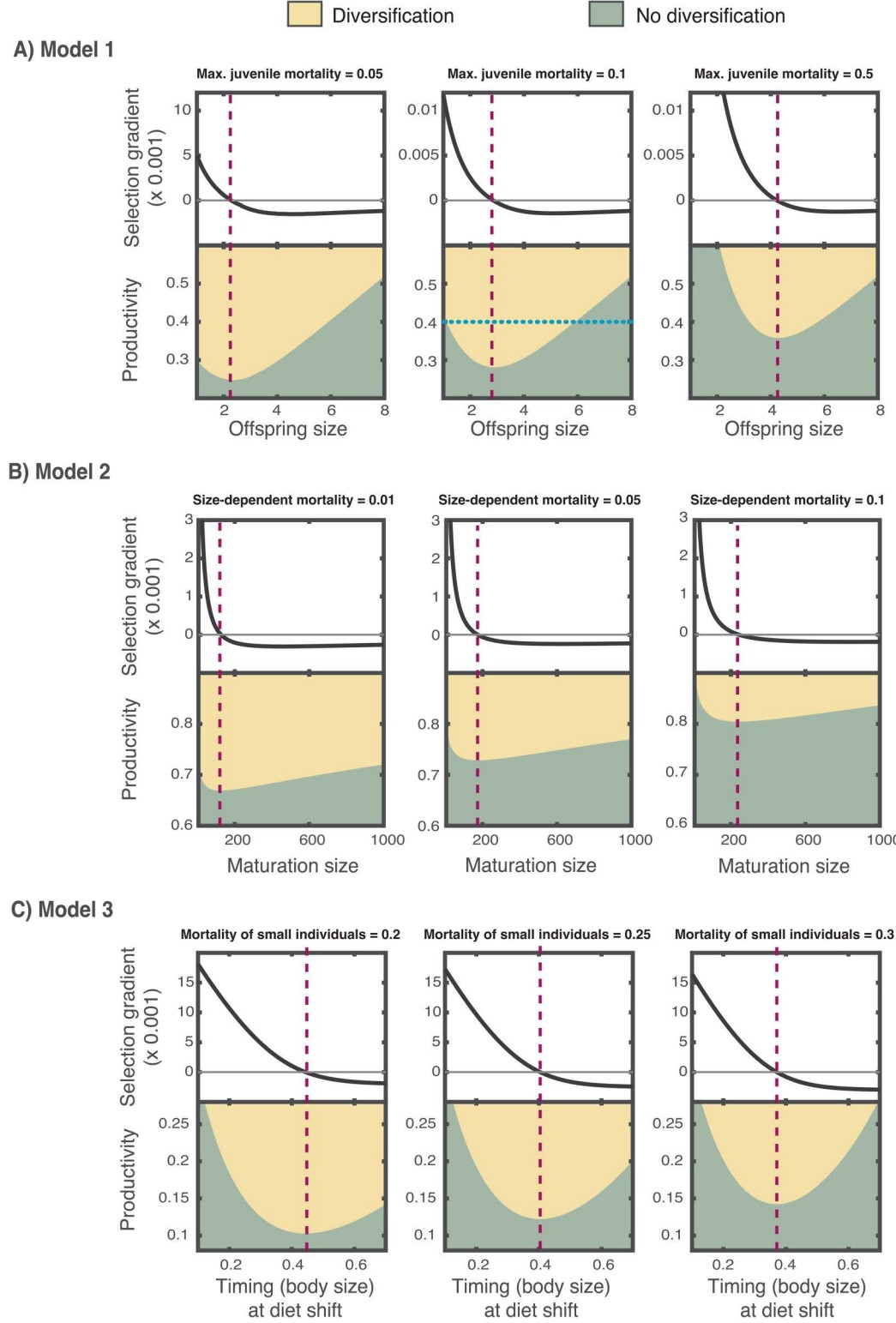

**Fig 4. Local adaptation in life history traits facilitates diversification.** The range of productivities over which diversification is enabled increases in the neighborhood of the optimal life history trait (i.e., where the selection gradient of the offspring size in A, maturation size in B, and timing at diet shift in C equals zero). For each model, three scenarios that differ in the mortality risk experienced by juveniles in A, all individuals in B, or the small individuals

in C were computed. The fitness landscape along the transect of the blue dotted line in A is shown in Fig 2G. Other parameter values of Model 1 as in Table A in S5 Text, of Model 2 as in Table B in S6 Text and of Model 3 as in Table C in S6 Text. The code needed to generate this Figure can be found in https://zenodo.org/records/17049771.

### General patterns: Optimal life histories maximize the potential for diversification

Across all studied models, we found that the productivity necessary for diversification decreases as the distance of the life history trait from its optimum becomes smaller (Fig 4). Extending our previous results from Model 1, the analysis of two additional model systems shows that life-history adaptation makes the environmental conditions for diversification, in this case, productivity, less restrictive. Notably, the three models differ substantially in their formulation of the population dynamics, the evolving life history traits, and the assumptions underlying life history processes (Fig 1B). Yet, despite these differences, they consistently demonstrate that natural selection, by driving life history adaptation, facilitates trophic diversification.

## Discussion

Here, we showed that life history adaptation results in stronger intraspecific competition for food resources, which in turn promotes diversification. Both theory [12,13] and empirical evidence [14–16] have demonstrated that by causing negative frequency-dependent interactions, intraspecific resource competition can be a source of disruptive selection. This may increase phenotypic diversity within a population [71] or result in ecologically driven sympatric speciation in case barriers to gene flow evolve between divergent phenotypes [72,73]. In either case, diversity is enhanced. Previous studies hypothesized that natural selection, by operating at the population level, may have consequences at the community level that ultimately promote diversity [74–76]. However, the underlying mechanisms remained unclear. Our results show that, after colonization of a new environment, life history evolution can facilitate trophic diversification, promoting diversity. Diversity, in turn, has been shown to influence dispersal, and thus colonization rates, by altering ecological interactions that exert selection on dispersal-related traits [77–82]. A self-reinforcing feedback loop promoting diversity may thus emerge in case the outcome of the altered interactions promotes dispersal (Fig 5). Our work suggests that natural selection operating on life history traits may be at the core of this feedback, emerging as a driver of self-organization in diverse ecological communities.

We find that a fast rate of adaptation of life history traits accelerates diversification, boosting lineage diversity. Empirical evidence from ray-finned fishes, the largest vertebrate clade, lends support to the validity of this prediction. In this clade, species richness across lineages and diversification rates correlate positively with the evolutionary rate of body size [83]. This morphological trait influences multiple life history traits [84–87], but it also plays a role in providing access to certain ecological niches. For example, the emergence of piscivorous species in various fish radiations entails an increase in body size [88–90]. Therefore, disentangling the effects of body size evolution on life history from its effects on ecological traits associated with niche specialization is challenging. Future research is needed to evaluate our prediction, e.g., by studying the evolutionary rates of life history traits, e.g., fecundity-at-age, maturation size, offspring size, etc., rather than body size, in related radiating and non-radiating taxa or taxonomic groups.

Our findings suggest that rapid local adaptation in life history traits may facilitate diversification. The case of Salmonids may qualitatively support this prediction. Various lineages of these fish have independently diversified producing an array of species exploiting diverse resources within temperate lakes when new habitats have become available (e.g., after the last glaciation) [90–92]. The extent of local adaptation in salmonids is remarkably high and manifests quickly following the invasion of a new habitat [93]. Additionally, life history traits in salmonids, such as offspring size, are documented to evolve rapidly in response to a shift in selection pressures (e.g., in the transition from wild to captivity, salmon populations quickly evolve small egg size due to the removal of selection imposed by predation) [94]. Our results may thus provide an explanation linking fast life history evolution, rapid local adaptation, and propensity to diversify.

   

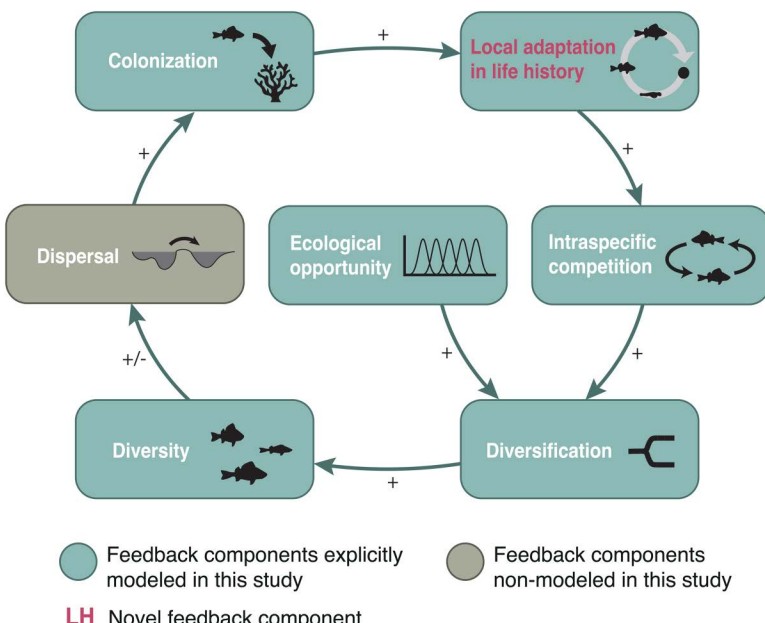

**Fig 5. An eco-evolutionary feedback loop promoting diversity.** Ecological diversification is common in complex landscapes composed of multiple ecosystems connected by dispersal (e.g., fish radiations across multiple lakes or bird radiations across island archipelagos). After colonization of a novel environment, local adaptation in life history results in stronger intraspecific competition, which, in the presence of ecological opportunity, induces disruptive selection. As a consequence, ecological diversification driven by niche specialization occurs, increasing phenotypic diversity within populations or species diversity in case barriers to gene flow evolve. In either case, functional diversity is enhanced, which, in turn, can positively influence dispersal and thereby the rate of colonization of novel environments. Although our model does not explicitly incorporate spatial structure, the evolutionary trajectories are assumed to begin from a colonization event, which is implicitly linked to dispersal across the landscape.

Here, we studied ecological diversification leading to specialization to exploit a variety of resources. Previous theory, in contrast, has usually modeled ecological diversification assuming a single resource (e.g., ref [95–97]), even when considering multiple evolving traits (e.g., ref [98,99]). This model feature makes it impossible for natural selection to drive a trait away from an evolutionary equilibrium at which selection turns disruptive. In other words, one of the conditions for diversification, more precisely, the condition for convergence stability [55], is always satisfied in those models. We show that life history traits affect diversification exactly via this condition. Therefore, by assuming a single food resource that inherently satisfies convergence stability, classical models of ecological diversification mask the effects of life history evolution on diversification. The assumption of a single food resource dominates existing theory on ecological diversification, despite empirical evidence showing that ecological diversification typically involves specialization to exploit diverse resources [10,100]. Only a few previous studies have considered multiple resources [27,73] and found that the condition for convergence stability is not always satisfied, as in our model. However, we are not aware of any models that have included evolution of life history traits in this context.

Our findings suggest that lineages with a higher rate of evolution of life history traits are expected to have higher diversification rates. We found this by numerically determining the evolutionary dynamics of lineages that differ in the mutation rate associated with the life history trait. However, other genetic processes may affect the rate at which life history adaptation occurs in response to natural selection. For instance, gene duplication, recombination, and epigenetic modification can also contribute to creating phenotypic variation that is both heritable and adaptive [101]. Therefore, by altering the adaptation rate of life history traits, these processes could have a similar effect on diversification as the mutation rate. Future research should consider explicit trait genetic architecture, using, e.g., individual-based model simulations [102], to investigate the role of these processes in life history evolution and its consequences for diversification.

Central to the mechanism promoting diversity introduced here is the feedback between the individual and its environment: the individuals' diet collectively affects the food resource availability, which in turn changes the profitability of an individual's diet. This feedback enables the emergence of a frequency-dependent selection regime, which is fundamental to inducing disruptive selection and thus diversification [12,13]. In this study, we examined the effect of life history evolution on diversification when the feedback between the individual and its environment results in the depletion of the food resources that provide ecological opportunity. In this scenario, we found that life history evolution facilitates diversification because the optimal life history strategy coincides with the strategy causing the strongest intraspecific competition for these resources. However, the utilization of additional resources may complicate diversification dynamics. This scenario can occur when only a part of the population has access to ecological opportunity, while the other part is limited by a different food resource. For example, in many pairs of sympatric species of fish [103] and anurans [104] with complex life cycles, only large individuals are specialized to feed on specific resources, whereas small individuals of multiple species share the same ecological niche. In such cases, if resource competition is stronger among small than among large individuals, small individuals grow slowly, resulting in low recruitment into the life stage that has access to ecological opportunity. As a consequence, competition for the resources that provide ecological opportunity is weak, hindering diversification [103]. In such scenario, life history adaptation is driven by the availability of the resources that provide ecological opportunity as well as another resource; therefore, the optimal life history strategy would not necessarily coincide with the strategy causing the strongest intraspecific competition for the resources providing ecological opportunity. Future research is needed to unveil the consequences of life history evolution for the diversification of organisms with complex life cycles with intraspecific competition for food in multiple life stages.

Our study is among the first to integrate life history theory with diversification models. This integrative approach proved essential for uncovering the role of life history evolution in facilitating trophic diversification. Because life-history traits shape fitness through their effects on survival and reproduction, their evolution alters population demography, which, in turn, impacts major ecological processes, such as intraspecific competition, as our study shows. Given that intraspecific competition is central to many ecological and evolutionary processes underlying biodiversity dynamics, including species coexistence, future work should explore the broader consequences of life-history evolution beyond ecological diversification. Understanding these consequences will require linking life history theory with other ecological and evolutionary theories. We have demonstrated how such integration unveils a general influence of life history evolution on diversification, highlighting the need to synthesize diverse research bodies to better understand the origin and maintenance of biodiversity.

## Supporting information

**S1 Text. Ecological dynamics and derivation of the fitness gradient and curvature of the fitness landscape.**
(PDF)

**S2 Text. Analytical derivation of the ecological equilibrium.**
(PDF)

**S3 Text. Analytical derivation of conditions that enable diversification.**
(PDF)

**S4 Text. Analytical proof that the minimum productivity required for diversification occurs when the life history trait is at its optimum.**
(PDF)

**S5 Text. Parameterization of Model 1. Table A.** Variables and parameters of Model 1.
(PDF)

**S6 Text. Description and parameterization of Model 2 and Model 3. Table B.** Life history parameters of Model 2 (in S6 Text). **Table C.** Life history parameters of the non-dimensionalized Model 3. **Figure A:** Energy allocation and body size at maturation.
(PDF)

## Acknowledgments

The authors thank Blake Matthews for helpful feedback on an early version of this manuscript.

## Author contributions

**Conceptualization:** P. Catalina Chaparro-Pedraza, Claudia Bank.

**Formal analysis:** P. Catalina Chaparro-Pedraza.

**Funding acquisition:** Claudia Bank.

**Investigation:** P. Catalina Chaparro-Pedraza.

**Methodology:** P. Catalina Chaparro-Pedraza.

**Resources:** Claudia Bank.

**Software:** P. Catalina Chaparro-Pedraza.

**Supervision:** Claudia Bank.

**Visualization:** P. Catalina Chaparro-Pedraza.

**Writing – original draft:** P. Catalina Chaparro-Pedraza.

**Writing – review & editing:** Claudia Bank.

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
