## [Editor Report · Decision Letter 0]

16 Apr 2025

Dear Dr Chaparro-Pedraza,

Thank you for submitting your manuscript entitled "Life history evolution facilitates trophic diversification" for consideration as a Research Article by PLOS Biology.

Your manuscript has now been evaluated by the PLOS Biology editorial staff, as well as by an academic editor with relevant expertise, and I'm writing to let you know that we would like to send your submission out for external peer review.

Once your full submission is complete, your paper will undergo a series of checks in preparation for peer review. After your manuscript has passed the checks it will be sent out for review. To provide the metadata for your submission, please Login to Editorial Manager (https://www.editorialmanager.com/pbiology) within two working days, i.e. by Apr 18 2025 11:59PM.

Kind regards,

Roli Roberts

Roland Roberts, PhD

Senior Editor

PLOS Biology

rroberts@plos.org

---

## [Decision Letter · Decision Letter 1]

15 Jul 2025

Dear Dr Chaparro-Pedraza,

Thank you for your patience while your manuscript "Life history evolution facilitates trophic diversification" went through peer-review at PLOS Biology. Your manuscript has now been evaluated by the PLOS Biology editors, an Academic Editor with relevant expertise, and by three independent reviewers.

You'll see that reviewer #1 thinks the research question is important and logical, and that the results are novel and interesting. His/her main concern is that the framing should make it clearer that there is some controversy about the breadth of conditions under which trait diversification can occur (there are some further minor textual requests). Reviewer #2 says that the results are “interesting and most probably correct” but wonders if they are somewhat unsurprising in the context of the significant literature on multitrait evolution; s/he suggests some modification to the framing to address this, and has a number of constructive textual and presentational requests. Reviewer #3 calls the study “timely, well-structured, and theoretically robust” but wants you to tone down some excessively general claims, unpack some model results for the general biological reader, and move some helpful material from the supplement to the main paper.

In light of the reviews, which you will find at the end of this email, we are pleased to offer you the opportunity to address the comments from the reviewers in a revision that we anticipate should not take you very long. We will then assess your revised manuscript and your response to the reviewers' comments with our Academic Editor aiming to avoid further rounds of peer-review, although we might need to consult with the reviewers, depending on the nature of the revisions.

IMPORTANT:

a) I'm wondering if you could make the very succinct Title a bit more explicit for our wider readership? I was thinking something like " "Life history evolution and natural selection drive biodiversity through feedback mechanisms" (this may be wrong, but it might give you an idea of what we're after!)

b) I should mention that we were not able to discuss the reviews with the Academic Editor in a timely fashion; the decision is clear, but it remains possible that the AE might have some further requests at the next stage.

c) Please address the concerns and requests from the reviewers.

d) Please provide a link to your funding body in the Financial Disclosure statement.

e) Please ensure that you comply with our Data and Code Policies; specifically, we need you to supply the code required to generate Figs 2ABCDEFG, 3, 4ABC, S1, either as a supplementary data file or as a permanent DOI’d deposition. I note that you already have a zipped supplementary folder containing the code; please can you confirm that this sufficient to generate the Figures? I also note that you have plans to deposit this is Zenodo; please do so and provide the URL/DOI (see below).

f) Please cite the location of the code clearly in all relevant main and supplementary Figure legends, e.g. “The code needed to generate this Figure can be found in S1 Code” or “The code needed to generate this Figure can be found in https://zenodo.org/records/XXXXXXXX

**IMPORTANT - SUBMITTING YOUR REVISION**

*Resubmission Checklist*

*Published Peer Review*

*PLOS Data Policy*

Sincerely,

Roli Roberts

Roland Roberts, PhD

Senior Editor

PLOS Biology

rroberts@plos.org

REVIEWS' COMMENTS:

Reviewer #1:

The authors of "Life history evolution facilitates trophic diversification" demonstrate that life history evolution can strengthen intraspecific competition, which in turn promotes diversification. The authors set up a very important, interesting, and logical question in the introduction (with one caveat described in the next paragraph). The models appear well formulated and are described with admirable clarity, especially given their complexity. The text is extremely well written with appropriate information constrained to the supplement, but no issue following along with only the main text. The results also appear novel and are interesting. The discussion makes empirical connections quite well and raises important caveats. There is quite a bit to like about this study and, while I made many comments, the majority of them are in service of improving clarity and should be relatively straightforward to address.

My biggest concern is that the introduction makes it sound as if sympatric ecological trait diversification is a relatively easy process that is well established empirically. I think that this is somewhat oversold. Indeed, while theory (cited here) suggests that such adaptive speciation is plausible, other work suggests that the conditions under which branching an occur is more restricted (e.g., Gavrilets 2005 Evolution or Buerger & Schneider 2006 Am Nat). While this does not require a dramatic reframing, the authors should make clearer that there is some controversy around these claims (either in the introduction or discussion). This comment particularly applies to lines 24 and 30-31. It is worth noting that the authors use more balanced language at the beginning of the discussion.

Specific comments:

Line 7: The phrase "structural features" is not clear to me.

Line 41: I see what the authors are getting at here, but I think the statement is a bit ambiguous. The traits controlling resource acquisition in past models are not necessarily subject to divergent selection. Indeed, under some conditions, selection remains stabilizing at equilibrium due to the assumption of a unimodal distribution of resources in the environment (e.g., Dieckmann & Doebeli 1999 Nature). This should be rephrased to be more accurate/nuanced.

Line 43: Typo, should say "strengthens".

Lines 78-79: A bit more explanation of why this is the appropriate measure of productivity in the model would be helpful.

Lines 86-87: It seems misleading to refer to this as a tradeoff because specialization on a single resource is only costly in the model (i.e., a narrower niche breadth does not confer a greater maximum attack rate but only reduces the attack rate on suboptimal resources). A similar comment applies on line 159.

Line 103: May be clearer as "that is, as the ratio . . ."

Lines 104-105: Would it be more appropriate to describe this as a birth rate?

Line 133: The term "simulated" is misleading in my opinion. This led me to believe the authors were considering individual-based simulations (as is often the case in models of adaptive speciation), rather than numerically integrating the differential equations in a deterministic model. Please rephrase to make this clearer.

Equation 3: It appears D is not defined in the main text

Lines 176-177: It may be worth explicitly pointing out that similar intuition also applies for why there exists a minimum productivity for diversification to occur.

Lines 180-181: This is a nice analytical result. It would be nice to add an explicit statement about what components of life history evolution affect condition 2 and in what direction. This can be pieced together by the reader from the information on line 167, but I think it would be more helpful to say it explicitly.

Figure 2 caption: "equilibriums" should instead be "equilibria".

Linen 205: I'm struggling to see how the sentence beginning with "As a consequence" follows from the previous sentence. Is it because, with so much resource depletion, selection favors trying to generalize and use both resources?

Lines 247: It would be useful to cite figure 1 in this section since you give an overview of these models there.

Lines 281-283: The focus on dispersal here and in figure 5 comes across as strange and tangential to the point of the study. I do not understand why dispersal is discussed alongside the modeling results given that it is not considered and that the results seem to stand on their own merits without this discussion of dispersal. I may be missing something, but I think some work is needed to better connect this text/figure to the modeling results.

Lines 326-347: A very important caveat to point out that is well described by the authors.

SI3 and SI4: "I" is used instead of "we" even though this is a multi-author paper.

Suggestions that the authors should feel free to ignore:

Line 14: I find the parenthetical more distracting than helpful. The phrase "environmental conditions" is clear on its own.

Line 134: I found it confusing to have figures 2 and 3 cited here, before results are presented.

Lines 212-219: I'm not seeing that this paragraph really adds any new information that is not obvious from the previous paragraph. I think it could be cut, or if there are new pieces, merged with the previous paragraph or made more explicit.

Reviewer #2:

The manuscript "Life history evolution facilitates trophic diversification" describes how the adaptation of developmental traits, such as juvenile body size and the age of switch to an adult diet, affect the propensity for evolutionary branching in another trait that defines preferences for various resources. Three models that are considered point to the conclusion that the two-trait coevolution results in diversification of the

trait related to resource preference. Some analytic results confirm and explain the outcomes of simulations.

In my opinion, the work is interesting and most probably correct. However, the conclusions come as no surprise to anybody familiar with the body of literature on multitrait evolution, of which the Authors are apparently unaware. For example, in

Ito, H. C., and U. Dieckmann. 2014. Evolutionary branching under slow directional evolution. Journal of Theoretical Biology 360:290-314. doi:10.1016/j .jtbi.2013.08.028.

Ispolatov, I., V. Madhok, and M. Doebeli. 2016. Individual-based models for adaptive diversification in high-dimensional phenotype spaces. Journal of Theoretical Biology 390:97-105.

It is studied under what conditions multidimensional phenotypes diversify "on the fly", not necessarily coming to a convergent stable strategy with zero selection gradient.

So the diversification of trait \eta in the direction orthogonal the ongoing adaptation of trait l (or other developmental traits in models 2 and 3) that happens for a certain interval of parameters does not come as something completely unexpected, but as yet another example of the fairly well-studied phenomenon.

So I suggest to tune down the emphasis on the novelty of interplay between the evolution of those two traits, but to rather focus on better linking the model to empirical data (e.g. making the distribution of food resources more realistic, explaining and justifying the functional forms and parameter intervals , showing the actual trajectories of trait evolution, etc).

Another general recommendation is to unify the terminology applied to evolution of l and \eta. After all, both are evolving under "natural selection" and in principle can undergo branching. The use of terms specific for the evolution of l and \eta creates unnecessary confusion as if two different processes are at play.

Below are several more specific suggestions:

Lines 29-31, In a realistic high-dimensional phenotype space, branching can occur on a fly without convergence to a fitness minimum, refs above

Line 32 Predation?

What is the difference between the "ecological diversification process" and "natural selection" (line 39)? Isn't the former just a specific case of the latter? I think the whole paragraph (35-45) is totally misleading.

Lines 97-98 The following phrase in not clear, "Hence, somatic growth only increases when the assimilated food exceeds the metabolic cost "

In the definition of Model 1 it has to be clearly stated that the juveniles don't actually grow, but rather stay as juveniles with the constant size l until maturing into adults at the rate specified by (101)

Lines 148-149 "For diversifica6on to occur, theory proposes that a population must experience disruptive selection for a significant amount of time (13)" What is changing with time?

Statements like "when the directional selection ceases" (line 137) are ambiguous as it is not clear to what directional selection, in l or \eta, they apply.

Lines 161-163 Here again, this has to be re-worded as the evolving phenotype is 2-dimensional and convergence to an attractor is not a necessary condition for diversification.

Line 188 What if the first scenario were started with l close to its diversification-promoting value (1.5, line 196?)?

Lines 220-235 This it a totally predictable outcome. In my opinion, its description should not constitute a dedicated subsection.

Lines 249-250 "In model 2, we study the effect of the evolution of body size at maturation, which determines the transition of energy allocation from growth to reproduction (figure S1). If it refers to ":w", why not to write it? I

Line 255. The definition of Model 3 should be way more quantitative.

Both models 2 and 3 should be properly defined and analyzed in the main text, or completely placed in the supplementary materials.

Does the size of juveniles actually change with time in models 2 and 3? If so, there should be equations that describe it, or a statement saying otherwise.

Lines 338-341. I cannot follow this argument. How the competition among juveniles is different from that among adults? With properly set parameters, the mechanism for diversification should work universally.

There should be more explanations to the figures. What do colours and black arrows mean in Fig. 1G? in the simulation described in Fig. 3, are both strains present at the beginning of simulation? It is hard to reconcile the fairly broad yellow areas in Fig. 4 with the statement that diversification is only happening when the selection gradient in \eta becomes zero. It would help to see the actual trajectories of various strains, possibly for both l and \eta coordinates.

In the Discussion, it is worth noting once again that the observed phenomena are a particular case of diversification in a two-dimensional trait space.

In Fig. 5, how the dispersal phase is related to diversity or diversification?

Reviewer #3:

The manuscript "Life history evolution facilitates trophic diversification" presents a compelling theoretical exploration of how life history evolution contributes to the emergence of ecological diversity. Using a suite of adaptive dynamics models, the authors demonstrate that natural selection acting on life history traits (such as offspring size, age at maturation, and timing of diet shift) enhances intraspecific competition, thereby creating conditions that facilitate niche diversification. Their models consistently show that life history adaptation reduces the environmental thresholds (e.g., productivity) needed for diversification and accelerates its occurrence. By integrating life history theory with ecological diversification models, the study provides a novel framework to explain how organismal traits shape biodiversity in a self-reinforcing feedback loop. This work is timely, well-structured, and theoretically robust, with potential to spark empirical research testing the macroevolutionary implications of life history evolution.

The following are areas for improvement:

1. Overextension of claims regarding empirical generality: While the theoretical results are strong, several claims about empirical implications (e.g., links between offspring size evolution and species richness) seem to be overstated. For example, references to salmonids and ray-finned fishes (lines 299-300) should be tempered, as these remain hypotheses without direct causal tests linking life history evolution to diversification.

2. Insufficient biological intuition: Some model descriptions, particularly in the results for Model 1 (lines 90-132), are highly technical and would benefit from additional biological interpretation. For example, the ecological meaning of compound parameters like E and G (line 166) could be clarified for a general audience.

3. Overreliance on SI: While mathematical derivations are expected in a theoretical biology paper, the heavy reliance on SI for core insights (e.g., conditions for diversification) may reduce accessibility. Including intuitive summaries of key equations (e.g., Eq. 3-4) and trade-off implications in the main text would strengthen the presentation.

Minor suggestions:

Line 16: "Our results call for a better integration…" - consider specifying which research fields should be integrated (e.g., "integrating empirical life history studies with theoretical models of niche diversification").

Line 46: Add a reference or brief explanation to support the claim that life history evolution alters intraspecific competition. Also, here the authors refer to "insights" (plural) but provide a singular reference.

Line 94-95: Clarify the biological significance of parameter γ scaling juvenile foraging relative to adults.

Line 107-110: The assumption that juvenile mortality decreases exponentially with offspring size should be justified with a brief biological rationale and a couple of references to the relevant literature.

Line 146: "…which we used to calculate the fitness landscape" - clarify how this is done in a more intuitive way, e.g., linking population structure to fitness gradients.

Line 178-179: "Life-history evolution… affects Condition 2" - better explain why only this condition (and not mutual invasibility) is altered by life history traits.

Line 221-223: Spell out how mutation rate differences were implemented—does this mimic variation in evolvability or genetic variance?

Line 259-260: Include more biological background on the trait "timing of diet shift" - e.g., which taxa exhibit it and how it links to ecological diversification.

Line 267: "The three models greatly differ…" - consider summarising in a short table or schematic the main features of Models 1-3 for clarity.

Line 296-297: "Our results provide an explanation linking fast life history evolution…" - a citation here is required to help ground this speculative bridge between theory and observation.

Line 310-313: "…preclude the possibility…" - this sentence could be clearer by explicitly stating that single-resource models inherently satisfy convergence stability and therefore mask life history effects.

Line 348-349: Consider explicitly stating that this study is among the first to formally integrate life history theory with diversification models.

---

## [Editor Report · Decision Letter 2]

29 Sep 2025

Dear Dr Chaparro-Pedraza,

Thank you for your patience while we considered your revised manuscript "Life history evolution facilitates trophic diversification" for publication as a Research Article at PLOS Biology. This revised version of your manuscript has been evaluated by the PLOS Biology editors and the Academic Editor.

Based on our Academic Editor's assessment of your revision, we are likely to accept this manuscript for publication, provided you satisfactorily address the following data and other policy-related requests (I had included these in my previous decision letter, but you may not have seen them?

IMPORTANT - please attend to the following:

a) I'm wondering if you could make the very succinct Title a bit more explicit for our wider readership? I was thinking something like " "Life history evolution and natural selection drive biodiversity through feedback mechanisms" (this may be wrong, but it might give you an idea of what we're after!)

b) Please provide a link to your funding body in the Financial Disclosure statement.

c) Please ensure that you comply with our Data and Code Policies; specifically, we need you to supply the code required to generate Figs 2ABCDEFG, 3, 4ABC, S1, either as a supplementary data file or as a permanent DOI’d deposition. I note that you mention a Zenodo deposition (https://zenodo.org/records/17049771) which contains a lot of Matlab code; please can you confirm that this is sufficient to generate the Figures?

d) Please cite the location of the code clearly in all relevant main and supplementary Figure legends, e.g. “The code needed to generate this Figure can be found in https://zenodo.org/records/17049771”.

We expect to receive your revised manuscript within two weeks.

*Published Peer Review History*

*Press*

Sincerely,

Roli Roberts

Roland Roberts, PhD

Senior Editor

rroberts@plos.org

PLOS Biology

DATA POLICY:

Regardless of the method selected, please ensure that you provide the individual numerical values that underlie the summary data displayed in the following figure panels as they are essential for readers to assess your analysis and to reproduce it: Figs 2ABCDEFG, 3, 4ABC, S1. NOTE: the numerical data provided should include all replicates AND the way in which the plotted mean and errors were derived (it should not present only the mean/average values).

CODE POLICY

Per journal policy, if you have generated any custom code during the course of this investigation, please make it available without restrictions. Please ensure that the code is sufficiently well documented and reusable, and that your Data Statement in the Editorial Manager submission system accurately describes where your code can be found. As the code that you have generated is important to support the conclusions of your manuscript, its deposition is required for acceptance.

DATA NOT SHOWN?

---

## [Editor Report · Decision Letter 3]

29 Oct 2025

Dear Dr Chaparro-Pedraza,

Thank you for the submission of your revised Research Article "Evolving life-history traits promote biodiversity via eco-evolutionary feedback mechanisms" for publication in PLOS Biology. On behalf of my colleagues and the Academic Editor, Jean-Michel Gaillard, I'm pleased to say that we can in principle accept your manuscript for publication, provided you address any remaining formatting and reporting issues. These will be detailed in an email you should receive within 2-3 business days from our colleagues in the journal operations team; no action is required from you until then. Please note that we will not be able to formally accept your manuscript and schedule it for publication until you have completed any requested changes.

Sincerely, 

Roli Roberts

Senior Editor

PLOS Biology

rroberts@plos.org